# Dairy Consumption at Breakfast among Southeast Asian Children: Associations with Nutrient Intake from the South East Asian Nutrition Surveys II (SEANUTS II)

**DOI:** 10.3390/nu16193229

**Published:** 2024-09-24

**Authors:** Nadja Mikulic, Ilse Khouw, Swee Ai Ng, Nipa Rojroongwasinkul, Nawarat Vongvimetee, Nga Thuy Tran, Van Khanh Tran, Rini Sekartini, Dian Novita Chandra, Bee Koon Poh, Jyh Eiin Wong, Cécile M. Singh-Povel, Nanda de Groot

**Affiliations:** 1FrieslandCampina, 3818 LE Amersfoort, The Netherlands; ilse.tan-khouw@frieslandcampina.com (I.K.); sweeai.ng@frieslandcampina.com (S.A.N.); cecile.singh-povel@frieslandcampina.com (C.M.S.-P.); nanda.degroot@frieslandcampina.com (N.d.G.); 2Institute of Nutrition, Mahidol University, Nakhon Pathom 73170, Thailand; nipa.roj@mahidol.ac.th (N.R.); nawarat.von@mahidol.ac.th (N.V.); 3National Institute of Nutrition, Hanoi 100000, Vietnam; thuynga1997@gmail.com (N.T.T.); trankhanhvan.nin@gmail.com (V.K.T.); 4Faculty of Medicine, Universitas Indonesia, Jakarta 10430, Indonesia; rsekartini@yahoo.com (R.S.); diannovitach@yahoo.com (D.N.C.); 5Faculty of Health Sciences, Universiti Kebangsaan Malaysia, Kuala Lumpur 50300, Malaysia; pbkoon@ukm.edu.my (B.K.P.); wjeiin@ukm.edu.my (J.E.W.)

**Keywords:** dairy, breakfast, children, nutrient intake, Southeast Asia

## Abstract

Background: Children’s rapid growth requires adequate nutrient intake, with breakfast, especially when incorporating dairy, playing an important role. This study examined associations between dairy consumption at breakfast and energy and nutrient intake among children in Southeast Asia. Methods: Utilizing the cross-sectional South East Asian Nutrition Surveys II dataset, using a 24 h dietary recall and questionnaires distributed among 10,286 children aged 2 to 12 years from Malaysia, Thailand, Indonesia, and Vietnam, we investigated the impact of dairy consumption at breakfast on nutrient intake and examined associations between breakfast consumption and the prevalence of stunting and overweight/obesity among 10,135 children. Results: Although most of the children consumed breakfast, only 37%, 27%, 16%, and 18% in Malaysia, Thailand, Indonesia, and Vietnam, respectively, included dairy in this meal, with younger children being significantly more likely to consume dairy at breakfast (*p* < 0.001). Dairy consumers had significantly higher intakes of vitamins A, B_12_, and D and calcium both at breakfast and throughout the day (*p* < 0.001). Breakfast skippers had a 29% increased risk of being overweight/obese. Conclusions: Dairy consumption at breakfast was associated with enhanced nutrient intake and overall diet quality among Southeast Asian children. The association between breakfast habits and anthropometric outcomes highlights the need to address nutritional deficiencies and promote healthy growth and development.

## 1. Introduction

Childhood nutrition is important for overall health and development, and breakfast plays a key role in providing essential nutrients for growth and is associated with a lower risk of overweight and obesity among children [1,2,3,4]. Additionally, research suggests that breakfast consumption positively impacts cognitive performance [5] and cardiometabolic factors [3]. Despite these benefits, previous studies conducted in Malaysia using food habit questionnaires found that 11.7% and 12.7% of children aged 6 to 17 years skipped or had irregular breakfasts, respectively [6], and that over one-third (36%) of children aged 6 to 12 years skipped breakfast at least once a week, which was associated with higher odds of overweight/obesity [7]. Notably, regular breakfast consumption was associated with healthier body weight statuses, highlighting the importance of establishing regular breakfast habits.

Similarly, in Indonesia, only 31.6% of children aged 2 to 12 years were classified as having adequate breakfast consumption practices [8], emphasizing the importance of comprehensively understanding breakfast habits in Southeast Asia (SEA), where malnutrition remains a significant concern [9]. Findings from the South East Asian Nutrition Surveys II (SEANUTS II) pertaining to Malaysia and Vietnam highlight concerns regarding stunting (~9%), overweight (~11%) and obesity (~10%) among children aged 6 months to 12 years, along with low micronutrient intake identified through a 24 h dietary recall, with most children failing to meet recommended nutrient intake guidelines, particularly for vitamin D (~96%) and calcium (~82%) [9,10]. Moreover, there is a high prevalence of vitamin D insufficiency (~26%) among those aged 4 to 12 years. Comparable results were observed in Thailand [11] and Indonesia (the corresponding data have been submitted for publication).

Insufficient dairy consumption, along with factors such as clothing style and outdoor activity, may contribute significantly to vitamin D deficiency [12]. Milk and dairy products are good sources of essential nutrients vital for growth and development, including energy, proteins, phosphorus, magnesium, vitamin D, and calcium [13,14]. A cluster analysis utilizing data from the National Health and Nutrition Examination Survey for children aged 2 to 18 years revealed that including dairy at breakfast was associated with a higher daily intake of vitamin D, calcium, and potassium [15]. Thus, a well-balanced breakfast, particularly one incorporating dairy, may play a significant role in fulfilling the nutritional requirements for the rapid biological and physical growth of children.

Utilizing the SEANUTS II dataset, our study aimed to assess the associations between dairy consumption at breakfast and energy and nutrient intake at breakfast and throughout the day among children aged 2 to 12 years in Malaysia, Thailand, Indonesia, and Vietnam using a 24 h dietary recall. Given the lack of prior research on the associations between breakfast consumption and stunting, as well as the limited data on breakfast consumption and overweight/obesity among children in SEA, our secondary objective was, using a child food habit questionnaire (CFHQ), to examine associations between breakfast consumption and the prevalence of stunting and overweight/obesity.

## 2. Materials and Methods

### 2.1. Study Design and Participants

This sub-study is part of SEANUTS II (*n* = 13,933), a multi-center, cross-sectional survey conducted from May 2019 to April 2021 in Malaysia (*n* = 2989), Thailand (*n* = 3478), Indonesia (*n* = 3465), and Vietnam (*n* = 4001) that aimed to gather comprehensive nutritional data from children aged 6 months to 12 years [16].

In this breakfast sub-study, the primary objective was to examine, separately in each of the four countries, the impact of dairy consumption during breakfast on energy and nutrient intake at breakfast and total daily energy and nutrient intake using a 24 h dietary recall. The secondary objective was to examine associations between breakfast consumption and the prevalence of stunting and overweight/obesity across all countries using a CFHQ. CFHQs were only collected from children aged 2 years and above. Therefore, the inclusion criteria for both objectives specified that only apparently healthy children aged 2 to 12 years, who were citizens of the studied countries and for whom valid data had been collected, were included in the 24 h dietary recall and CFHQ. Exclusion criteria for both objectives were signs of physical disability; genetic, cardiovascular, or respiratory illness that limited physical activity; medical history or illness within the past 3 months; feeling unwell on the day of measurement; and implausible energy intake determined from the 24 h dietary recall. Recruitment was limited to one child per household. Additionally, for the secondary objective, exclusion criteria involved incomplete CFHQ data collection and extreme values for height-for-age *z*-score (HAZ, <−6 or >6), weight-for-age *z*-score (WAZ, <−5 or >5), weight-for-height *z*-score (<−6 or >5), and body mass index (BMI)-for-age *z*-score (BAZ, <−5 or >5).

### 2.2. Sociodemographic Characteristics and Anthropometric Statuses

The methodology employed to gather sociodemographic characteristics and anthropometric statuses, including body weight, height, and BMI, is explained in the paper outlining the rationale and study design of SEANUTS II [16]. For children under 5 years, WAZ, HAZ, BAZ, and weight-for-age z-scores were computed using the World Health Organization (WHO)’s Anthro software (version 3.2.2 in Malaysia, Thailand, and Vietnam and version 3.1.0 in Indonesia). Meanwhile, for children aged 5 years and above, WAZ, HAZ, and BAZ were calculated using the WHO’s AnthroPlus software (version 1.0.4 in Malaysia and Vietnam, and version 1.0.3 in Thailand and Indonesia). Definitions for stunting, overweight, and obesity were based on WHO criteria [17,18].

### 2.3. Dietary Intake Assessment Using a 24-h Dietary Recall and Child Food Habit Questionnaire

For the primary objective, we used a one-day 24 h dietary recall, and for the secondary objective, we used a one-week CFHQ, whose methodologies are described in the paper outlining the rationale and study design of SEANUTS II [16]. Dietary supplements were excluded from the dietary analysis of this sub-study.

To identify and exclude implausible energy intake data from the 24 h dietary recall in Malaysia, Indonesia, and Vietnam, we computed the ratio of reported EI to predicted energy expenditure (EE) [19]. EE was estimated by multiplying basal metabolic rate [20] by a moderate physical activity level [21]. Subsequently, we applied a 99% confidence interval (CI), accounting for an 8.2% within-subject variation in EE and a 23% within-subject variation in EI [22]. Only participants with an EI:EE ratio falling within 0.27–1.73 were included in the analysis. In Thailand, implausible EI data were assessed using the predicted total EI method [23] based on the Food and Agriculture Organization’s recommendations for a moderate physical activity level [24]. The standard deviation (SD) was derived from the coefficient of variation of the reported EI and the predicted total EI within the study population [24]. Plausible CI limits were defined as the predicted total EI ± 2 SD.

### 2.4. Definition of Breakfast, Dairy, and Other Food Groups

In the 24 h dietary recall, breakfast was defined as the first eating occasion following overnight sleep consumed before 12 p.m., excluding water, tea, and coffee without milk. Conversely, in the CFHQ, breakfast was defined based on the caregiver’s interpretation. Food items consumed at breakfast were categorized into five food groups according to local dietary guidelines (cereals and grains, vegetables, fruits, dairy, and meat and proteinaceous foods) [25,26,27,28] and extra foods, which encompassed food items not categorized within the other food groups [29]. Dairy encompassed liquid and powdered animal-based dairy products, excluding human breast milk and plant-based dairy alternatives. Evaporated milk, condensed milk, butter, cream, ice cream, and sour milk were not considered dairy. Standardized serving sizes for dairy products were established according to regional recommendations. In Malaysia, one serving of milk was defined as 250 mL, one serving of milk powder was defined as 30 g of powder, one serving of yoghurt was defined as 270 g, one serving of Greek yoghurt was defined as 135 g, and one serving of cheese was defined as 40 g [25]. In Thailand, one serving of milk equated to 200 mL, and one serving equated to 150 g and 30 g for yogurt and cheese, respectively [26]. Since specific recommendations for powdered milk were lacking in Thailand, Malaysia’s serving size guidelines for milk powder were adopted. For Indonesia, one serving of milk was defined as 200 mL, while one serving for milk powder, goat milk, and cheese was defined as 20 g (powder), 185 mL, and 40 g, respectively [27]. In Vietnam, one serving comprised 100 mL for milk, 15 g for milk powder, 100 g for yoghurt, and 15 g for cheese [28]. Further details on serving sizes for other food items and daily food group recommendations can be found in the following reports [25,26,27,28].

### 2.5. Sample Size Calculation and Statistical Analysis

The sample size calculation for this study is explained in the paper outlining the rationale and study design of SEANUTS II [16].

Statistical analyses were conducted using the R statistical programming environment (version 4.3.0). We used the pastecs package for descriptive statistics. To check assumptions, data normality in each country and subgroup was assessed using the Shapiro–Wilk W test. Homogeneity of variance was checked using Levene’s test from the car package. Data were visualized using the ggplot2 package. Results are presented as means ± SD for normally distributed data and as medians (interquartile range [IQR]) for non-normally distributed data. Chi-square tests were performed using the gmodels package to compare frequencies in categorical data if expected frequencies were above 5; otherwise, Fisher’s exact tests were conducted. In cases where the overall test for more than two groups was significant, standardized residuals were calculated to identify which groups contributed to the significant result. To make comparisons between two groups with continuous variables as outcomes, independent Welch’s *t*-tests were conducted for normally distributed data, while Wilcoxon’s rank-sum tests were used for non-normally distributed data. Analysis of variance and Tukey post hoc tests (corrected for number of tests) were employed for comparisons between more than two groups if assumptions were met (using the multcomp package); otherwise, Kruskal–Wallis tests with post hoc tests (which were corrected for the number of tests and for which the pgirmess package was used) were used. Outliers, defined as 1.5 times the IQR less than the first quartile or greater than the third quartile, were removed if they significantly influenced the outcome of the statistical test. *p* Values < 0.05 were considered statistically significant.

For the primary objective, we analyzed each country’s data separately and stratified the data by age group (2 to 3 years, 4 to 6 years, and 7 to 12 years) sex (female and male), residential area (urban and rural), and income group (high-income group, middle-income group, low-income group, and poor group). Income groups were defined using data from reputable sources such as the World Bank (Washington, DC, USA), Pew Research (Washington, DC, USA), Nielsen (New York, NY, USA), and Kantar (London, UK) [30].

For the secondary objective, we pooled the data from all four countries and conducted a multivariate binary logistic regression analysis. Initially, we estimated the crude odds ratios (OR) and their corresponding 95% CIs using univariate logistic regression. Subsequently, multivariate logistic regression analysis was performed to control for potential confounders. Residuals were examined to identify outliers, and those with a significant influence on the model were removed. We developed two separate models where the outcome variables were stunting and overweight/obesity (binary yes/no). The predictor variable was breakfast frequency, categorized from the CFHQ as skippers (0 to 2 days per week), semi-skippers (3 to 4 days per week), and non-skippers (5 to 7 days per week) [6]. Age group, sex, residential area, and income group were included as potential confounders. Confounders were added to the regression model if the model’s deviance was significantly different, which was indicated by a lower Akaike information criterion value. Additionally, data analysis was stratified by age group (2 to 6 years and 7 to 12 years) and the four countries.

## 3. Results

### 3.1. Participants

For the primary objective, we included 2438 children in Malaysia, 2643 children in Thailand, 2216 children in Indonesia, and 2989 children in Vietnam, and for the secondary objective, we included 10,153 children from all four countries (Figure 1).

### 3.2. Breakfast Intake According to the 24-Hour Dietary Recall

According to the 24 h dietary recall, 98% of the children consumed breakfast in Malaysia, 99% consumed it in Thailand and Indonesia, and 97% consumed it in Vietnam. Table 1 presents the sociodemographic characteristics of the children who consumed breakfast on the day of the 24 h dietary recall. In all countries, younger children consumed breakfast less frequently than older children (*p* < 0.001), and there were no differences in breakfast consumption frequency between female and male children. In Malaysia and Indonesia, a higher proportion of children in urban areas consumed breakfast compared to children in rural areas (*p* < 0.001 and *p* < 0.05, respectively), whereas the opposite trend was observed in Thailand and Vietnam (*p* < 0.001). In Malaysia, the highest proportion of children who consumed breakfast came from the high- and low-income groups, while the lowest proportion came from the middle-income and poor groups (*p* < 0.001). In Thailand and Indonesia, the highest proportion of children who consumed breakfast came from the low-income and poor groups, and the lowest proportion came from the high- and middle-income groups (*p* < 0.001). In Vietnam, the highest proportion of children who consumed breakfast came from the middle-income group, while the lowest proportion came from the high-income and poor groups (*p* < 0.001).

### 3.3. Consumption of Dairy and Other Food Groups According to the 24-Hour Dietary Recall

In Malaysia, Thailand, Indonesia, and Vietnam, 36.8%, 26.6%, 15.8%, and 17.5%, respectively, of breakfast consumers consumed dairy at breakfast. Table 2 illustrates that in all the analyzed countries, fewer children consumed dairy at breakfast as they get older, with the highest proportion of children who consumed dairy observed in the youngest age group and the lowest in the oldest age group (*p* < 0.001). In Malaysia and Indonesia, a higher proportion of children living in urban areas consumed dairy at breakfast compared to children living in rural areas (*p* < 0.001), whereas there was no difference observed in Thailand and Vietnam. Additionally, there was no difference in dairy consumption at breakfast between female and male children in all countries. Generally, a higher proportion of children from the high- and middle-income groups consumed dairy at breakfast than children from the low-income and poverty groups (*p* < 0.01 in Thailand and *p* < 0.001 in rest).

Table 3 illustrates the proportions of children who consumed various food groups at breakfast. In Malaysia and Indonesia, the extra food group emerged as the most frequently consumed, while in Thailand and Vietnam, cereals and grains were the predominant choice. Dairy products ranked as the third most consumed food group in Malaysia, the fourth in Thailand and Indonesia, and the fifth in Vietnam.

The median (IQR) dairy intake at breakfast among children who consumed dairy at breakfast is 0.9 (0.6) servings in Malaysia, contributing ~56% to the daily dairy intake. In Thailand, it is 1.0 (0.1) serving, contributing ~38% to the daily dairy intake. In Indonesia, the median (IQR) intake is 1.0 (0.8) serving, contributing ~29% to the daily diary intake, while in Vietnam, it is 1.1 (0.7) servings, contributing ~41% to the daily dairy intake. The recommended number of daily dairy servings is two in Malaysia [25] and Thailand [26], one serving in Indonesia [27], and four servings for children aged 1 to 9 years and six servings for children aged 10 to 12 years in Vietnam [28]. In Malaysia, Thailand, Indonesia, and Vietnam, only 17%, 23%, 24%, and 8%, respectively, of children who did and did not consume breakfast met these recommendations.

The total daily dairy intakes and breakfast dairy intakes categorized by country, age group, sex, residential area, and income group for children who consumed dairy at breakfast are shown in Figure 2. In Malaysia, the dairy intakes at breakfast were significantly different between age groups (*p* < 0.001), with the median (IQR) dairy intake being higher among children aged 2 to 3 years and 4 to 6 years compared to those aged 7 to 12 years (0.9 [0.5] and 1.0 [0.7], respectively, versus 0.8 [0.5] servings). Conversely, in Vietnam, dairy intakes at breakfast were significantly different between age groups (*p* < 0.001), with the oldest age group consuming more dairy than children aged 2 to 3 years and 4 to 6 years (1.8 [0.7] versus 1.1 [0.7] and 1.1 [0.7] servings, respectively). Furthermore, in Malaysia and Thailand, the total daily dairy intakes were significantly different between age groups (*p* < 0.001), with the median (IQR) intake being highest among children aged 2 to 3 years compared to children aged 4 to 6 years and 7 to 12 years (2.9 [2.2] versus 1.9 [2.2] versus 1.0 [0.9] servings, respectively, and 3.2 [2.4] versus 2.0 [1.2] versus 1.8 [1.2] servings, respectively). Similarly, in Vietnam, the total daily dairy intakes were significantly different between children aged 2 to 3 years and children aged 4 to 6 years and 7 to 12 years (*p* < 0.001; 3.2 [2.1] versus 2.3 [2.0] and 1.8 [1.8] servings, respectively), whereas in Indonesia, the dairy intakes were significantly different between age groups, with children aged 2 to 6 years and 4 to 6 years having higher intakes than those aged 7 to 12 years (3.8 [3.3] and 4.0 [3.6] versus 1.5 [1.3] servings, respectively).

In Malaysia and Vietnam, the median (IQR) dairy intake at breakfast was higher among male children compared to that among female children (0.9 [0.6] versus 0.8 [0.6] servings, *p* < 0.05, and 1.5 [0.7] versus 1.1 [0.7] servings, *p* < 0.01, respectively). In Malaysia, total daily dairy intake was higher among male children compared to that among female children (1.7 [2.1] versus 1.4 [2.0] servings, *p* < 0.01).

Additionally, in Malaysia, the median (IQR) dairy intake at breakfast was higher in urban areas than in rural areas (0.9 [0.6] versus 0.8 [0.6] servings, *p* < 0.05), whereas there were no area differences in total daily dairy intake in all countries. Moreover, no differences were observed in total daily dairy intake and intake at breakfast between income groups in all countries, except in Thailand, where the total daily dairy intake was lower in the poor group than in the low-income group (2.6 [1.8] versus 2.0 [2.0] servings, *p* < 0.05).

### 3.4. Energy and Nutrient Intake at Breakfast between Children Consuming versus Not Consuming Dairy at Breakfast According to the 24-h Dietary Recall

Table 4 presents the energy and nutrient intakes at breakfast among children who consumed dairy versus those who did not. In Malaysia, children who consumed dairy at breakfast had lower energy and carbohydrate intakes compared to those who did not consume dairy (*p* < 0.01). Conversely, in Thailand, Indonesia, and Vietnam, energy intake was higher among dairy-consuming children (*p* < 0.001). In Indonesia, the inclusion of dairy resulted in higher carbohydrate, docosahexaenoic acid (DHA), and choline intakes (*p* < 0.001). Protein and fat intakes were higher when dairy was included at breakfast in all countries except Malaysia (*p* < 0.001). In Malaysia, fiber intake was higher when dairy was included at breakfast (*p* < 0.001). Children consuming dairy at breakfast had higher intakes of vitamin A, vitamin B_12_, vitamin C (~2–3-fold), vitamin D (~5–30×), and calcium (~3–6×) at breakfast compared to those not consuming dairy in all countries (*p* < 0.001). In Malaysia, Thailand, and Indonesia, including dairy at breakfast led to higher intakes of vitamin B_1_ and vitamin B_2_ (*p* < 0.001). In Malaysia, β-carotene data were ~four times higher when dairy was consumed at breakfast (*p* < 0.001). Vitamin B_3_ levels were higher in Malaysia (*p* < 0.001), but lower in Thailand (*p* < 0.01), when dairy was consumed at breakfast. Iron intake was higher among children who consumed dairy at breakfast in Malaysia and Indonesia (*p* < 0.001), but no differences were observed in Thailand and Vietnam. Similarly, zinc intake was higher in Indonesia and Vietnam among children who consumed dairy at breakfast (*p* < 0.001). In Malaysia and Thailand, sodium intake was lower among children who consumed dairy at breakfast (*p* < 0.001 and *p* < 0.01, respectively), while potassium and phosphorus intakes were higher (*p* < 0.001).

Given the notable variations observed across the different age groups, as depicted in Figure 2, we conducted a detailed analysis within this subgroup (Appendix A).

In Malaysia (Appendix A), upon performing age stratification, energy intake ceased to show significant differences between children who consumed dairy at breakfast and those who did not across all age groups. Protein intake was not different between children who consumed and did not consume dairy at breakfast in the younger age groups, while among children aged 7 to 12 years who consumed dairy at breakfast, the intake was higher (*p* < 0.01). Conversely, carbohydrate intake was lower among children who consumed dairy at breakfast compared to non-consumers (*p* < 0.05). Across all age groups, fat intake did not differ between the two groups. The intakes of all vitamins and minerals were higher when dairy was consumed at breakfast compared to when dairy was not consumed at breakfast in all age groups (*p* < 0.001), except for sodium intake, which was lower (*p* < 0.001).

In Thailand (Appendix A), energy intake was only higher among children who consumed dairy at breakfast compared to those who did not consume dairy at breakfast in the younger age groups (*p* < 0.001). Protein intake was higher across all age groups (*p* < 0.001, and *p* < 0.01 among those 7 to 12 years old). Following age stratification, only within the 2 to 3 years age group was carbohydrate intake higher among children who consumed dairy at breakfast compared to non-consumers (*p* < 0.01), while no differences were observed in the other age groups. Furthermore, across all age groups, children who consumed dairy at breakfast had higher intakes of fat; fiber; vitamins A, B_1_, B_2_, B_12_, and D; calcium; potassium; and phosphorus (*p* < 0.001, and *p* < 0.01 for vitamin B_1_ and potassium among children aged 7 to 12 years). Pre-stratification analysis revealed lower vitamin B_3_ intakes among children who consumed dairy at breakfast, yet among children aged 2 to 3 years, intake was higher (*p* < 0.001), with no differences in the other age groups. Similarly, pre-stratification analysis indicated there was a higher vitamin C intake among children who consumed dairy at breakfast; however, this difference persisted only in the youngest age group (*p* < 0.001) and ceased in the older ones. Iron intake did not differ among all children, but after stratifying by age group, intake was higher when dairy was consumed at breakfast in the youngest age group (*p* < 0.001), contrasting with its being lower in the oldest age group (*p* < 0.01). Although zinc and magnesium intakes did not differ across all children, stratifying by age group revealed higher intake levels in the youngest age group when dairy was consumed at breakfast (*p* < 0.001 and *p* < 0.05, respectively). After stratification, sodium intake was not different when dairy was consumed at breakfast among all children, in contrast to the lower intake observed when dairy was consumed at breakfast pre-stratification.

In Indonesia (Appendix A), energy, macro- and micronutrient, vitamin, and mineral intakes were higher among children who consumed dairy at breakfast across all age groups (*p* < 0.001, except *p* < 0.05 for fat intake among children aged 7 to 12 years). After age stratification, only within the 4 to 6 years age group was choline intake higher among children who consumed dairy at breakfast compared to non-consumers (*p* < 0.001), while no differences were observed in the other age groups. Similarly, pre-stratification, DHA intake was higher among children who consumed dairy at breakfast; however, stratification by age group revealed that intake was higher only in the younger age groups (*p* < 0.001).

In Vietnam (Appendix A), energy, protein, fat, calcium, and zinc intakes and those for all vitamins were higher among children who consumed dairy at breakfast across all age groups (*p* < 0.001). Although carbohydrate and iron intakes were not different between children who consumed and did not consume dairy at breakfast among all children, stratification by age group revealed higher intakes among those 4 to 6 years old (*p* < 0.05) and 2 to 3 years old (*p* < 0.05), respectively, who consumed dairy at breakfast.

### 3.5. Total Daily Energy and Nutrient Intakes among Children Who Consumed versus Did Not Consume Dairy at Breakfast According to the 24-h Dietary Recall

Table 5 shows the total daily energy and nutrient intakes among children who consumed versus did not consume dairy at breakfast. In Thailand, energy intakes were lower among children who consumed dairy at breakfast compared to those among non-consumers (*p* < 0.05), whereas in Indonesia, intakes were higher (*p* < 0.001). In Thailand and Vietnam, carbohydrate intakes were lower when dairy was consumed at breakfast (*p* < 0.001 and *p* < 0.05, respectively), while in Indonesia, carbohydrate, choline, and DHA intakes were higher (*p* < 0.001, *p* < 0.01, and *p* < 0.001, respectively). Protein intakes did not differ between the two groups, except in Indonesia, where consumers of dairy at breakfast had higher intakes (*p* < 0.001). In Thailand, Indonesia, and Vietnam, fat intakes were higher when dairy was included at breakfast (*p* < 0.001). In Thailand and Indonesia, fiber intakes were higher when dairy was included at breakfast (*p* < 0.001). In all countries, children who consumed dairy at breakfast had higher intakes of vitamin A, vitamin B_12_, vitamin D, and calcium compared to children who did not consume dairy at breakfast (*p* < 0.001). In Malaysia and Indonesia, vitamin C intakes were higher among children who consumed dairy at breakfast compared to those among children who did not consume dairy at breakfast (*p* < 0.001), whereas there were no differences in intake between these two groups in Thailand and Vietnam. In Malaysia, Thailand, and Indonesia, including dairy at breakfast led to higher intakes of vitamin B_1_ (*p* < 0.001, *p* < 0.01, and *p* < 0.001, respectively) and vitamin B_2_ (*p* < 0.001). In Malaysia, β-carotene intakes were higher when dairy was consumed at breakfast (*p* < 0.001). In Malaysia, vitamin B_3_ intakes were higher when dairy was included at breakfast (*p* < 0.001), whereas they were lower in Thailand (*p* < 0.01). In Malaysia and Indonesia, children who consumed dairy at breakfast had higher iron intakes than children who did not consume dairy at breakfast (*p* < 0.001); however, in Thailand, iron intakes were lower among children who consumed dairy at breakfast (*p* < 0.01), and there were no differences in Vietnam between these two groups. In Indonesia, children who consumed dairy at breakfast had higher zinc intakes than children who did not consume dairy at breakfast (*p* < 0.001). Magnesium intake data were only available in Thailand, and the results showed that including dairy at breakfast led to a lower intake (*p* < 0.001). In Malaysia and Thailand, compared to children who did not consume dairy at breakfast, sodium intakes were lower among children who consumed dairy at breakfast (*p* < 0.001), and intakes of potassium (*p* < 0.001 and *p* < 0.05, respectively) and phosphorus were higher among children who consumed dairy at breakfast (*p* < 0.001).

Appendix A display the total daily energy and nutrient intakes among children who consumed breakfast, stratified by age group in Malaysia, Thailand, Indonesia, and Vietnam.

In Malaysia (Appendix A), energy and carbohydrate intakes were higher among children who consumed dairy at breakfast compared to those who did not consume dairy at breakfast among children aged 4 to 6 years (*p* < 0.001). Protein intakes were higher among children who consumed dairy at breakfast among children aged 2 to 3 years and 4 to 6 years (*p* < 0.05 and *p* < 0.001, respectively), while there were no differences among children aged 7 to 12 years. Fat intake was higher in older age groups among those who consumed dairy at breakfast compared to that among non-consumers (*p* < 0.05). The intakes of all vitamins and minerals were higher when dairy was consumed at breakfast compared to when dairy was not consumed at breakfast in all age groups (*p* < 0.001, except β-carotene in all age groups and vitamin B_12_ among children aged 7 to 12 years *p* < 0.01), except for sodium intake, which showed no difference between the groups.

In Thailand (Appendix A), energy intake was higher among children who consumed dairy at breakfast compared to those who did not consume dairy at breakfast, while post-stratification, energy intake was higher in the youngest age group (*p* < 0.05). Upon age stratification, carbohydrate, vitamin C, and vitamin B_3_ intakes were not different between children who consumed and did not consume dairy at breakfast across all age groups, whereas protein (*p* < 0.001 in the youngest age group and *p* < 0.05 in the older age groups); vitamin A (*p* < 0.001), B_2_ (*p* < 0.001), B_12_ (*p* < 0.001 in the younger age groups and *p* < 0.01 in the oldest age group), and D (*p* < 0.001); calcium (*p* < 0.001); and phosphorus (*p* < 0.001 in the youngest and oldest age groups and *p* < 0.01 among children aged 4 to 6 years) intakes were higher among children who consumed dairy at breakfast. Fat intake was higher among children who consumed dairy at breakfast across all age groups (*p* < 0.01, and *p* < 0.001 among children aged 2 to 3 years). Among children aged 4 to 6 years and 7 to 12 years, fiber (*p* < 0.05 and *p* < 0.01, respectively) and vitamin B_1_ (*p* < 0.001 and *p* < 0.01, respectively) intakes were higher among children who consumed dairy at breakfast. Before stratification by age group, iron intake was lower among children who consumed dairy at breakfast, while post-stratification, no difference was observed across the age groups. Zinc intake did not differ among all children, but after stratifying by age group, it was higher when dairy was consumed at breakfast in the youngest age group (*p* < 0.01). Similarly, pre-stratification analysis revealed lower magnesium intake among children who consumed dairy at breakfast; however, this difference persisted only among children aged 4 to 6 years (*p* < 0.05) and ceased for the other ones. Similarly, potassium intake was lower among all children who consumed dairy at breakfast; however, this difference persisted only among children aged 2 to 6 years (*p* < 0.01). After stratification, sodium intakes were not different when dairy was consumed at breakfast among all children, in contrast to the lower intake observed when dairy was consumed at breakfast pre-stratification.

In Indonesia (Appendix A), children who consumed dairy at breakfast had higher energy, protein, carbohydrate, fat, calcium, iron, and zinc intakes and intakes of all vitamins across all age groups (*p* < 0.001, and *p* < 0.01 for fat intake among children aged 7 to 12 years). Pre-stratification, fiber, choline, and DHA intakes were higher among children who consumed dairy at breakfast compared to those who did not; however, after stratification, these differences were only observed in the youngest age groups (fiber, *p* < 0.001; choline among children aged 2 to 3 years, *p* < 0.05, and among children aged 4 to 6 years, *p* < 0.001; and DHA among children aged 2 to 3 years, *p* < 0.001, and among children aged 4 to 6 years, *p* < 0.01).

In Vietnam (Appendix A), energy and protein intakes did not differ between children who consumed and did not consume dairy at breakfast among all children; however, upon stratifying by age group, intakes were higher among children who consumed dairy at breakfast in the older age groups (*p* < 0.01 and *p* < 0.05, respectively). Although carbohydrate intake throughout the day was lower among children who consumed dairy at breakfast among all children, this difference disappeared upon stratifying by age group. Fat (*p* < 0.001), vitamin A (*p* < 0.001), and vitamin B_12_ (*p* < 0.001, except *p* < 0.01 among children aged 2 to 3 years) intakes throughout the day were higher among children who consumed dairy at breakfast across all age groups. Vitamin C intakes did not differ between the two groups across all age groups. Vitamin D and calcium intakes were higher among children who consumed dairy at breakfast across all age groups (*p* < 0.001). Although iron and zinc intakes did not differ between children who consumed and did not consume dairy at breakfast among all children, stratification by age group revealed higher iron intake among children aged 7 to 12 years and higher zinc intake among children aged 4 to 6 years (*p* < 0.05) who consumed dairy at breakfast.

### 3.6. Association Analysis between Breakfast Skipping and Stunting and Overweight/Obesity

Among the 10,153 children across all four countries, 5.6% were identified as breakfast skippers, 4.6% were determined to be semi-skippers, and 89.8% were identified as non-skippers (Table 6). Older children (7 to 12 years old) were more likely to skip breakfast (73.8%), while younger children (2 to 6 years old) were more likely to be non-skippers (53.3%). Females were more prevalent among skippers (58.5%) and non-skippers (50.4%), whereas semi-skippers were more likely to be male (52.2%). Living in an urban residence was more common among skippers (53.7%) and semi-skippers (52.2%) compared to non-skippers (43.4%). The highest proportions of skippers and semi-skippers were from the low-income (40.2% and 36.9%, respectively) and poor groups (24.6% and 29.7%, respectively), while non-skippers were represented in the highest proportions in the low- (35.7%) and middle-income groups (28.4%). Examining country-specific trends, Malaysia and Indonesia stood out with the highest prevalence of breakfast skippers (50.7% and 27.1%, respectively) and semi-skippers (43.9% and 25.9%, respectively), while Vietnam and Thailand had the highest prevalence of non-skippers (31.7% and 26.2%, respectively). Notably, there is a significant association between breakfast-skipping behavior and the prevalence of overweight/obesity (*p* < 0.001). The proportion of overweight/obese children was highest among skippers (27.1%) and lowest among semi-skippers (18.5%). There were no differences between the three groups in terms of stunting.

We investigated the associations between breakfast frequency and stunting, as well as overweight/obesity, using multivariate binary logistic regression. The models controlled for various potential confounders, including age group, sex, residential area, and income group. The final models included 10,107 children, with 46 observations excluded due to missing data in the income group. The regression coefficients, their standard errors, and odds ratios with their 95% CIs are presented in Table 7.

Neither semi-skippers nor skippers showed a statistically significant association with stunting when compared to non-skippers. However, children aged 7 to 12 years old were significantly less likely (33%) to be stunted compared to younger children (OR = 0.67, 95% CI = 0.59–0.77, *p* < 0.001). Children living in rural areas were 26% more likely to be stunted than those in urban areas (OR = 1.26, 95% CI = 1.10–1.44, *p* < 0.01). The odds of stunting increased with the decrease in income levels: for the middle-income group (OR = 1.81, 95% CI = 1.38–2.40, and *p* < 0.001), for the low-income group (OR = 2.38, 95% CI = 1.84–3.11, and *p* < 0.001), and for the poor group (OR = 4.16, 95% CI = 3.20–5.78, and *p* < 0.001) compared to the highest-income group.

Skipping breakfast was significantly associated with an increased risk of being overweight/obese compared to eating breakfast (OR = 1.29, 95% CI = 1.05–1.58, *p* < 0.05). Children aged 7 to 12 years were significantly more likely to be overweight/obese than those aged 2 to 6 years (OR = 3.23, 95% CI = 2.90–3.60, *p* < 0.001). Males had greater odds of being overweight/obese compared to females (OR = 1.43, 95% CI = 1.29–1.59, *p* < 0.001). Lower income levels were associated with decreased odds of being overweight/obese: for the middle income group (OR = 0.86, 95% CI = 0.74–0.99, *p* < 0.05), for the low income group (OR = 0.66, 95% CI = 0.57–0.76, *p* < 0.001), and for the poor group (OR = 0.43, 95% CI = 0.36–0.51, *p* < 0.001) compared to the highest income group.

After stratification by age group and country (Appendix A), breakfast skippers aged 2 to 6 years had a significantly lower (60% lower) risk of being stunted compared to non-skippers (OR = 0.40, 95% CI = 0.18–0.76, *p* < 0.05), while no significant association was found between breakfast frequency and overweight/obesity. Among children aged 7 to 12 years, no significant association was found between breakfast frequency and stunting. However, breakfast skippers aged 7 to 12 years had a significantly higher (38%) risk of being overweight/obese compared to non-skippers (OR = 1.35, 95% CI = 1.08–1.69, *p* < 0.01).

Skipping breakfast was significantly associated with an 83% greater risk of being overweight compared to eating breakfast among Malaysian children (OR = 1.83, 95% CI = 1.38–2.42, *p* < 0.001). In Thailand and Indonesia, skipping breakfast was not significantly associated with overweight/obesity. For Vietnam, skipping breakfast was significantly associated with a 77% decreased risk of overweight/obesity compared to eating breakfast (OR = 0.23, 95% CI = 0.05–0.66, *p* < 0.05). Notably, skipping breakfast was not significantly associated with stunting when all four countries were analyzed separately.

## 4. Discussion

Our study aimed to explore the impact of dairy consumption at breakfast on energy and nutrient intake at breakfast and throughout the day among children aged 2 to 12 years in Malaysia, Thailand, Vietnam, and Indonesia. Additionally, we investigated potential associations between breakfast consumption and anthropometric outcomes such as overweight/obesity and stunting. According to the 24 h dietary recall, in Malaysia, Thailand, Indonesia, and Vietnam, only 37%, 27%, 16%, and 18% of the analyzed children, respectively, consumed dairy at breakfast, and our findings show that this varied among (1) countries, with Malaysia having the highest proportion of dairy consumers; (2) age groups, with the youngest age group having the highest proportion of dairy consumers; and (3) income groups, with the high- and middle-income groups having the highest proportions. Including dairy at breakfast was associated with higher intakes of fat; vitamins A, B_1_, B_2_, B_12_, and D; calcium; potassium; and phosphorous and lower intakes of sodium at breakfast and throughout the day in most countries. Further, children who skipped breakfast had a 29% greater risk of being overweight/obese compared to children who did not skip breakfast.

Dairy intake at breakfast contributed significantly to the total daily dairy intake, although most children did not meet the recommended daily dairy servings. For instance, only 17%, 23%, 24%, and 8% of children in Malaysia, Thailand, Indonesia, and Vietnam, respectively, met the recommendations. This indicates a widespread deficiency in dairy intake among children, highlighting the need for public health initiatives to promote dairy consumption to meet nutritional guidelines. Notably, in the oldest age group, dairy consumed at breakfast often constituted most of the total daily dairy intake. The total daily dairy intake varied significantly with age. In all four countries, younger children had the highest daily dairy intake, which declined with age. The shift away from dairy as children grow older may be influenced by changing dietary patterns or the availability of alternative food choices. The substitution of milk with alternative beverages has variously been attributed to increased autonomy in beverage choice and the availability of other beverages [31]. In general, Asian diets are relatively low in dairy foods compared to Western countries. The primary forms of dairy consumed in these regions are limited to liquid milk and milk powder, while the intake of butter and cheese remains low due to dietary habits [32]. In SEA, dairy products are not traditionally produced or consumed, which is further influenced by the relatively high cost of milk in these areas.

Associations between dairy and nutrient intake have been examined mainly in Western countries, showing that dairy intake is linked to higher intakes of numerous vitamins and minerals, including folate, vitamin B_6_, and vitamin B_12_ [33,34]. Similar to our results, studies from Japan [35] and the US [36] reported that milk consumption was associated with better macro- and micronutrient intakes among children and adults. Our analysis demonstrated that including dairy at breakfast was associated with a more favorable nutrient intake profile at breakfast. Specifically, across most countries, it was related to higher intakes of energy by 13–31%, protein by 23–30%, fat by 15–110%, vitamin A by >360%, vitamin B_1_ by 47–214%, vitamin B_2_ by 54–158%, vitamin B_12_ by 63–564%, vitamin C by >30%, vitamin D by >426%, calcium by >200%, potassium by 36–82%, and phosphorous by 49–68%. Additionally, it was associated with a 9–42% lower intake of sodium. This positive impact extends beyond breakfast, contributing to enhancing the nutrient profiles in daily diets by elevating levels of fat by 6–19%, vitamin A by 28–90%, vitamin B_1_ by 7–84%, vitamin B_2_ by 29–93%, vitamin B_12_ by 27–44%, vitamin D by 63–343%, calcium by 61–162%, potassium by 6–27%, and phosphorous by 14–26% and lowering sodium intake by 10–11% across most countries. This is particularly important given our previous research, which highlighted insufficient intakes of calcium and vitamin D in this region [9,10,11].

A Japanese study highlighted that calcium represented the greatest difference in inadequate daily intakes of nutrients between dairy and non-dairy consumers [35]. In contrast, our study showed that the highest difference in daily intake between children who consumed and did not consume dairy at breakfast was in relation to vitamin D, followed by calcium and vitamin A. At breakfast, the highest difference in nutrient intake was found to correspond to vitamin A, followed by vitamin D, vitamin C, calcium, and vitamin B_12_. Previous studies also mention a higher prevalence of excess daily saturated fat intake [35,36]. Our study found a higher intake of fat both at breakfast and throughout the day when dairy was included at breakfast in most countries. However, we did not differentiate between saturated and unsaturated fat intakes, which limits our ability to compare the impacts of full-fat versus low-fat dairy products.

Our age-stratified analysis revealed that the benefits of dairy consumption at breakfast were more pronounced among younger children, particularly those aged 2 to 6 years, among which we observed higher intakes of essential vitamins and minerals. Older children, while still benefiting from increased intakes of certain nutrients, showed less consistent patterns, highlighting the need for a continued emphasis on balanced breakfast choices as children age. The variations observed across different countries underscore the importance of considering regional dietary habits and the availability of food products when designing nutritional interventions. While dairy consumption at breakfast significantly increased energy and macronutrient intakes in Thailand, Indonesia, and Vietnam, this effect was not as pronounced in Malaysia. This suggests that cultural dietary practices and local food environments play crucial roles.

Including dairy in children’s diets improves diet quality and nutrient intake at breakfast and throughout the day. Our study, along with numerous others, demonstrates favorable associations between dairy intake and an increased intake of key nutrients, leading to improved diet quality and micronutrient status, particularly in relation to B-vitamins and vitamin D, among children and adults [37,38,39]. Because dairy products are rich in essential nutrients such as calcium, vitamin D, protein, and various other micronutrients crucial for children’s growth and development, several studies have highlighted their positive impact on growth parameters, and evidence suggests that dairy consumption may be inversely associated with the risk of developing overweight and obesity among children [40,41,42]. The potential mechanisms through which dairy may influence growth and weight include the regulation of appetite and satiety, the improvement of body composition, and the provision of bioactive compounds that support metabolic health [42,43]. A recent study highlighted the health benefits of replacing carbohydrate-rich breakfast components with one serving of dairy [44]. This change improved postprandial amino acid availability, glycemic control, and bone metabolism through increasing serum calcium. Adding a second serving of dairy augments postprandial amino acid and glucagon-like peptide-1 concentrations while further promoting satiety. Notably, including dairy at breakfast does not increase the intake of daily protein. It is noteworthy that disparities in energy and nutrient intake existed among the analyzed countries, emphasizing the need for tailored nutritional interventions based on regional dietary patterns.

Our multivariate analysis indicates that age, sex, residential area, and income level are significant predictors of stunting and overweight/obesity among children, while breakfast frequency does not show a significant direct association with stunting. These findings suggest that socioeconomic factors and geographic location play critical roles in influencing stunting. However, our study identified a concerning association between breakfast skipping and overweight/obesity. Children who skipped breakfast had a 29% higher risk of being overweight, reinforcing the importance of regular breakfast consumption in weight management. The results of a meta-analysis showed that skipping breakfast was associated with a higher risk of being overweight or obese among children and adolescents [45]. A potential explanation might be that skipping breakfast may lead to increased hunger later in the day, resulting in overeating or choosing unhealthy, calorie-dense foods [4,6]. Furthermore, delaying the timing of one’s first meal can contribute to body fat accumulation by disrupting biological processes such as clock-controlled gene regulation, satiety hormones, insulin modulation, and lipid metabolism [3,46], ultimately leading to body fat accumulation [47]. The high prevalence of breakfast skippers in Malaysia is a notable concern, as this group exhibited the highest percentage of overweight or obese children. The reasons for breakfast skipping may be lack of appetite, limited time, and unfamiliarity with breakfast habits, which might be partly linked to the early commencement of school hours [48]. Our findings are in line with other Malaysian cross-sectional studies, which found a positive association between breakfast skipping and overweight among children aged 6 to 17 years [6] and 6 to 12 years [7]. On the contrary, in Vietnam, skipping breakfast was associated with a decreased risk of becoming overweight. This unexpected finding requires further exploration to understand the cultural, dietary, and lifestyle factors contributing to this association in the Vietnamese context. However, our results may be related to the very small number of overweight/obese breakfast skippers in Vietnam, so they should be interpreted with caution.

We observed an interaction between skipping breakfast and the children’s ages concerning stunting. While skipping breakfast was significantly associated with a decreased risk of stunting among young children, it was associated with an increased risk of being overweight or obese among older children. These findings align with a cross-sectional study conducted in Hong Kong, which reported higher BMI *z*-scores for older children aged 9 to 18 years who skipped breakfast [45]. Although breakfast has long been considered the most important meal of the day, the authors suggest that overweight children, perhaps with the support of their parents, may consciously skip breakfast to reduce food consumption. However, regular breakfast consumption could have genuine health benefits, such as reducing satiety later in the day, which can prevent overeating and the consumption of more energy-dense, unhealthy foods [49]. This age-dependent relationship underscores the complexity of the nutritional impact of breakfast habits and emphasizes the importance of considering developmental stages when formulating public health strategies. Furthermore, our results and those of previous studies showed that breakfast skipping is less common among younger children but increases with age [50,51,52]. Younger children are usually closely monitored by their caregivers, which might ensure more regular and balanced meal patterns [53].

This study has several strengths, including its large sample size, multi-country design, and use of rigorous dietary assessment methods. Additionally, our investigation focused on countries in SEA, providing new insights into this region, whereas most previous studies examined other regions. However, certain limitations should be acknowledged, including the reliance on self-reported dietary data, which may be subject to recall bias. Further, the 24 h dietary recall was only conducted for one day instead of at least three days, inducing variation between weekdays and weekend days. Additionally, the cross-sectional nature of this study limits our ability to establish causality. We also did not adjust associations for physical activity levels and sleep, which might be associated with breakfast skipping and overweight or obesity. Despite these limitations, our findings underscore the need for comprehensive nutritional interventions, with a focus on promoting regular and high-quality breakfast consumption including dairy, especially for vulnerable populations in low-income households.

## 5. Conclusions

Our study provides valuable insights into childhood nutrition in SEA, emphasizing the important role of breakfast, particularly dairy in regard to consumption, in meeting essential nutrient requirements. Our findings reveal significant variations in dairy consumption at breakfast across countries, age groups, and income levels, with Malaysia, accounting for the youngest age group, and higher-income groups showing the highest proportions of dairy consumers. Despite the important contribution of dairy products to the diets of children, our results indicate a decline in dairy consumption and a tendency for decreasing intake with age. Further, including dairy at breakfast was associated with higher intakes of fat, vitamin A, most B-vitamins, vitamin D, calcium, potassium, and phosphorous and a lower intake of sodium both at breakfast and throughout the day. Additionally, children who skipped breakfast had a 29% greater risk of being overweight or obese compared to those who did not skip breakfast. These results highlight the importance of promoting regular breakfast consumption and incorporating dairy products into children’s diets to address nutritional deficiencies and support healthy growth and development among children in the SEA region. Public policies, such as implementing school milk programs, can help provide children with regular access to dairy and promote balanced breakfasts. Further research and longitudinal studies are warranted to validate these findings and support evidence-based interventions for improving childhood nutrition in diverse sociodemographic settings.

## Figures and Tables

**Figure 1 nutrients-16-03229-f001:**
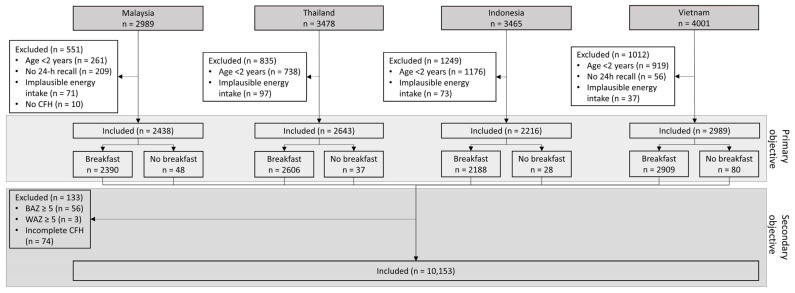
Participant flow chart concerning the primary objective, examining, separately in each of the four countries, the impact of dairy consumption during breakfast on energy and nutrient intake at breakfast and throughout the day using a 24 h dietary recall, and the secondary objective, examining associations between breakfast consumption and the prevalence of stunting and overweight/obesity across all countries using a child food habit questionnaire.

**Figure 2 nutrients-16-03229-f002:**
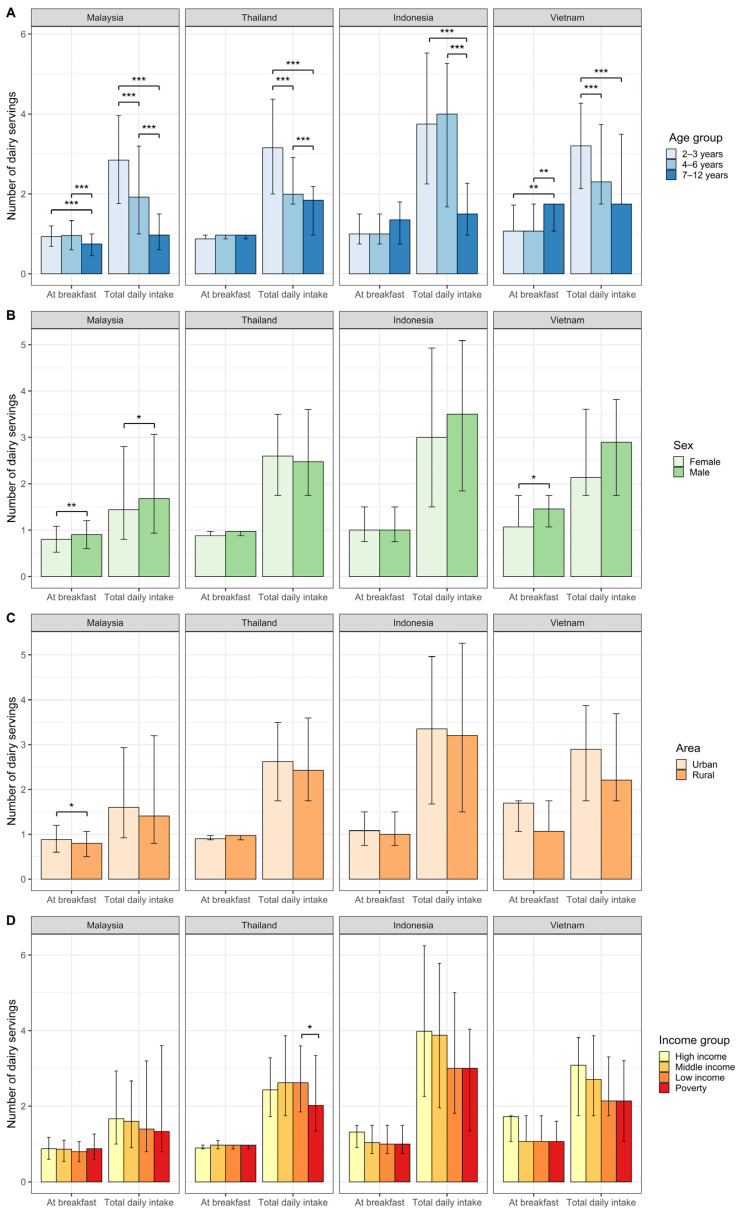
Median (interquartile range [IQR]) number of servings of total daily dairy and dairy at breakfast consumed by children who consumed dairy at breakfast per country (Malaysia: *n* = 880, Thailand: *n* = 692, Indonesia: *n* = 345, Vietnam: *n* = 508), categorized by (**A**) age group, (**B**) sex, (**C**) area, and (**D**) income group. We assessed group differences in number of servings using Wilcoxon’s rank-sum tests and Kruskal–Wallis tests with post hoc tests (corrected for number of tests). The bars indicate the median, and the error bars represent the IQR. Asterisks indicate significant differences based on *p* values (*** *p* < 0.001, ** *p* < 0.01, * *p* < 0.05).

**Table 1 nutrients-16-03229-t001:** Sociodemographic characteristics of children who consumed breakfast on the day of the 24 h dietary recall.

	Malaysia (*n* = 2390)	*p*-Value	Thailand (*n* = 2606)	*p*-Value	Indonesia (*n* = 2118)	*p*-Value	Vietnam (*n* = 2909)	*p*-Value
Age, *n* (%)								
2–3 years	322 (13.5%)	<0.001	722 (27.7%)	<0.001	627 (28.7%)	<0.001	662 (22.8%)	<0.001
4–6 years	751 (31.4%)		842 (32.3%)		521 (23.8%)		733 (25.2%)	
7–12 years	1317 (55.1%)		1045 (40.1%)		1040 (47.5%)		1514 (52.0%)	
Sex, *n* (%)								
Female	1232 (51.5%)	0.13	1292 (49.6%)	0.67	1131 (51.7%)	0.11	1447 (49.7%)	0.78
Male	1158 (48.5%)		1314 (50.4%)		1057 (48.3%)		1462 (50.3%)	
Area, *n* (%)								
Urban	1665 (69.7%)	<0.001	816 (31.3%)	<0.001	1152 (52.7%)	<0.05	871 (29.9%)	<0.001
Rural	725 (30.3%)		1790 (68.7%)		1036 (47.3%)		2038 (70.1%)	
Income group, *n* (%)								
High-income	840 (35.1%)	<0.001	252 (9.7%)	<0.001	41 (1.9%)	<0.001	508 (17.5%)	<0.001
Middle income	430 (18.0%)		296 (11.4%)		381 (17.4%)		1624 (55.8%)	
Low-income	728 (30.5%)		1297 (49.8%)		992 (45.3%)		623 (21.4%)	
Poor	349 (14.6%)		760 (29.2%)		774 (35.4%)		154 (5.3%)	
Missing values	43 (1.8%)		1 (0.0%)		0 (0%)		0 (0%)	

Data show frequencies and proportions (%). Chi-square tests were conducted within each stratification group and country.

**Table 2 nutrients-16-03229-t002:** Proportions of children who consumed and did not consume dairy at breakfast on the day of the 24 h dietary recall.

	Malaysia			Thailand			Indonesia			Vietnam		
	Did Not Consume Dairy (*n* = 1510)	Consumed Dairy (*n* = 880)	*p*-Value	Did Not Consume Dairy (*n* = 1917)	Consumed Dairy (*n* = 692)	*p*-Value	Did Not Consume Dairy (*n* = 1843)	Consumed Dairy (*n* = 345)	*p*-Value	Did Not Consume Dairy (*n* = 2401)	Consumed Dairy (*n* = 508)	*p*-Value
Age, *n* (%)												
2–3 years	106 (32.9%)	216 (67.1%)	<0.001	398 (55.1%)	324 (45.1%)	<0.001	440 (70.2%)	187 (29.8%)	<0.001	493 (74.5%)	169 (25.5%)	<0.001
4–6 years	406 (54.1%)	345 (45.9%)		635 (75.4%)	207 (24.6%)		428 (82.2%)	93 (17.8%)		563 (76.8%)	170 (23.2%)	
7–12 years	998 (75.8%)	319 (24.2%)		884 (84.6%)	161 (15.4%)		975 (93.8%)	65 (6.3%)		1345 (88.8%)	169 (11.2%)	
Sex, *n* (%)												
Female	801 (65.0%)	431 (35.0%)	0.05	928 (71.8%)	364 (28.2%)	0.06	958 (84.7%)	173 (15.3%)	0.53	1184 (81.8%)	263 (18.2%)	0.31
Male	709 (61.2%)	449 (38.8%)		986 (75.0%)	328 (25.0%)		885 (83.7%)	172 (16.3%)		1217 (83.2%)	245 (16.8%)	
Area, *n* (%)												
Urban	1002 (60.2%)	663 (39.8%)	<0.001	611 (74.9%)	205 (25.1%)	0.26	931 (80.8%)	221 (19.2%)	<0.001	717 (82.3%)	154 (17.7%)	0.84
Rural	508 (70.1%)	217 (29.9%)		1303 (72.8%)	487 (27.2%)		912 (88.0%)	124 (12.0%)		1684 (82.6%)	354 (17.4%)	
Income group, *n* (%)												
High-income	471 (56.1%)	369 (43.9%)	<0.001	165 (65.5%)	87 (34.5%)	<0.01	27 (65.9%)	14 (34.1%)	<0.001	395 (77.8%)	113 (22.2%)	<0.001
Middle income	271 (63.0%)	159 (37.0%)		206 (73.7%)	90 (30.4%)		291 (76.4%)	90 (23.6%)		1308 (80.5%)	316 (19.5%)	
Low-income	475 (65.2%)	253 (34.8%)		956 (73.7%)	341 (26.3%)		821 (82.8%)	171 (17.2%)		555 (89.1%)	68 (10.9%)	
Poor	264 (75.6%)	85 (24.4%)		586 (77.1%)	174 (22.9%)		704 (91.0%)	70 (9.0%)		143 (92.9%)	11 (7.1%)	
Missing values	29 (67.4%)	14 (32.6%)		1 (100.0%)	0 (0.0%)		0 (0.0%)	0 (0.0%)		0 (0.0%)	0 (0.0%)	

Data show frequencies and proportions (%). Chi-square tests were conducted within each stratification group and country.

**Table 3 nutrients-16-03229-t003:** Proportions of children who consumed food groups at breakfast on the day of the 24 h dietary recall.

	Malaysia (*n* = 2390)	Thailand (*n* = 2606)	Indonesia (*n* = 2188)	Vietnam (*n* = 2909)
Cereal/Grains, *n* (%)	1561 (65.3%)	2371 (91.0%)	1964 (89.8%)	2539 (87.3%)
Meat and Proteinaceous Foods, *n* (%)	776 (32.5%)	2258 (86.6%)	1602 (73.2%)	1921 (66.0%)
Dairy, *n* (%)	880 (36.8%)	692 (26.6%)	345 (15.8%)	508 (17.5%)
Fruits, *n* (%)	95 (4.0%)	111 (4.3%)	62 (2.8%)	81 (2.8%)
Vegetables, *n* (%)	282 (11.8%)	641 (24.6%)	563 (25.7%)	579 (19.9%)
Extra Foods, *n* (%)	1585 (66.3%)	1414 (54.3%)	2066 (94.4%)	1279 (44.0%)

Data show frequencies and proportions (%).

**Table 4 nutrients-16-03229-t004:** Energy and nutrient intakes at breakfast among children who consumed versus those who did not consume dairy at breakfast recorded using the 24 h dietary recall.

	Malaysia		Thailand		Indonesia		Vietnam	
	Did Not Consume Dairy (*n* = 1510)	Consumed Dairy (*n* = 880)	Did Not Consume Dairy (*n* = 1917)	Consumed Dairy (*n* = 692)	Did Not Consume Dairy (*n* = 1843)	Consumed Dairy (*n* = 345)	Did Not Consume Dairy (*n* = 2401)	Consumed Dairy (*n* = 508)
Energy, kcal	262 (219)	240 (186) **	326 (248)	369 (240) ***	235 (164)	309 (183) ***	207 (160)	264 (169) ***
Protein, g	7.3 (9.1)	7.5 (6.6)	13.0 (10.2)	16.0 (10.7) ***	8.2 (6.4)	10.7 (6.8) ***	7.5 (7.0)	9.7 (7.4) ***
Carbohydrates, g	36.8 (29.4)	32.4 (26.1) ***	42.5 (36.0)	41.5 (29.9)	28.2 (22.0)	37.4 (25.4) ***	31.4 (24.6)	33.1 (23.3)
Fat, g	8.2 (9.3)	7.8 (7.2)	9.4 (12.1)	13.4 (10.7) ***	10.6 (9.9)	12.2 (10.4) ***	4.1 (6.1)	8.6 (6.4) ***
Fiber, g	-	-	0.66 (0.85)	1.16 (1.63) ***	0.55 (0.90)	0.94 (1.12) ***	-	-
Vitamin A, µg RAE	**-**	-	29 (128)	129 (138) ***	-	-	2 (47)	93 (62) ***
Vitamin A, µg RE	105 (149)	168 (133) ***	-	-	35 (76)	148 (114) ***	-	-
β-carotene, µg	9.1 (68.8)	40.4 (53.9) ***	-	-	-	-	-	-
Vitamin B_1_, mg	0.17 (0.27)	0.30 (0.38) ***	0.15 (0.27)	0.22 (0.40) ***	0.07 (0.08)	0.22 (0.20) ***	-	-
Vitamin B_2_, mg	0.26 (0.29)	0.40 (0.34) ***	0.19 (0.25)	0.49 (0.25) ***	0.14 (0.16)	0.36 (0.29) ***	-	-
Vitamin B_3_, mg ^1^	2.30 (2.67)	2.75 (2.69) ***	2.45 (3.15)	2.14 (2.94) **	-	-	-	-
Vitamin B_12_, µg	0.22 (0.60)	0.50 (0.46) ***	0.36 (0.59)	0.79 (0.82) ***	0.43 (0.65)	0.70 (0.72) ***	0.11 (0.39)	0.73 (0.39) ***
Vitamin C, mg	4.8 (10.8)	17.3 (17.9) ***	0.7 (5.5)	1.7 (8.7) ***	0.0 (1.4)	13.0 (12.7) ***	0.0 (0.6)	1.3 (0.9) ***
Vitamin D, µg	0.40 (1.23)	2.40 (2.47) ***	0.38 (1.29)	2.00 (1.44) ***	0.25 (0.65)	2.64 (2.18) ***	0.06 (0.26)	1.80 (1.11) ***
Calcium, mg	84 (123)	252 (164) ***	34 (63)	229 (85) ***	55 (58)	232 (195) ***	26 (34)	173 (91) ***
Iron, mg	2.13 (2.29)	3.12 (2.56) ***	1.47 (1.36)	1.47 (1.86)	1.50 (1.47)	2.82 (2.41) ***	1.15 (1.09)	1.22 (1.23)
Zinc, mg	-	-	1.09 (0.95)	1.14 (1.20)	0.90 (0.66)	1.90 (1.32) ***	1.06 (1.13)	1.28 (1.12) ***
Magnesium, mg ^2^	-	-	18.2 (18.3)	18.2 (20.4)	-	-	-	-
Sodium, mg	286 (452)	166 (231) ***	426 (538)	386 (469) **	-	-	-	-
Potassium, mg	148 (168)	269 (174) ***	192 (180)	261 (260) ***	-	-	-	-
Phosphorus, mg	111 (111)	186 (123) ***	125 (125)	186 (163) ***	-	-	-	-
Choline, mg	-	-	-	-	33 (138)	41 (94) ***	-	-
DHA, mg	-	-	-	-	3.3 (19.7)	9.1 (18.9) ***	-	-

Data are presented as medians (IQR). Data were analyzed with Wilcoxon’s rank-sum tests. Significant differences from children who did not consume dairy are indicated as *** *p* < 0.001, and ** *p* < 0.01. ^1^ Outliers removed in Thailand: did not consume dairy (*n* = 1856) and consumed dairy (*n* = 646). ^2^ Outliers removed in Thailand: did not consume dairy (*n* = 1779) and consumed dairy (*n* = 671).

**Table 5 nutrients-16-03229-t005:** Energy and nutrient intakes throughout the day among children who consumed and did not consume dairy at breakfast according to the 24 h dietary recall.

	Malaysia		Thailand		Indonesia		Vietnam	
	Did Not Consume Dairy (*n* = 1510)	Consumed Dairy (*n* = 880)	Did Not Consume Dairy (*n* = 1917)	Consumed Dairy (*n* = 692)	Did Not Consume Dairy (n = 1843)	Consumed Dairy (*n* = 345)	Did Not Consume Dairy (*n* = 2401)	Consumed Dairy (*n* = 508)
Energy, kcal	1380 (636)	1380 (548)	1480 (660)	1420 (581) *	1120 (602)	1290 (583) ***	1100 (526)	1130 (522)
Protein, g	50.3 (27.5)	50.8 (23.0)	54.7 (27.6)	55.9 (25.8)	34.4 (19.8)	41.1 (20.4) ***	47.3 (26.3)	47.9 (23.3)
Carbohydrates, g	186 (94)	184 (80)	192 (95)	172 (88) ***	141 (80)	165 (80) ***	167 (80)	162 (74) *
Fat, g	45.8 (27.3)	47.1 (24.6)	52.1 (30.2)	55.3 (28.5) ***	43.9 (28.0)	48.9 (28.1) ***	25.4 (19.2)	30.3 (18.3) ***
Fiber, g	-	-	5.65 (3.99)	5.81 (4.71)	3.95 (4.17)	4.89 (4.52) ***	-	-
Vitamin A, µg RAE	-	-	298 (292)	437 (325) ***	-	-	271 (344)	363 (288) ***
Vitamin A, µg RE	570 (499)	732 (481) ***	-	-	317 (392)	603 (504) ***	-	-
β-carotene, µg	633 (1350)	779 (1410) ***	-	-	-	-	-	-
Vitamin B_1_, mg	0.90 (0.70)	1.19 (0.79) ***	0.81 (0.72)	0.87 (0.79) **	0.49 (0.47)	0.90 (0.61) ***	-	-
Vitamin B_2_, mg	1.17 (0.82)	1.51 (0.90) ***	1.13 (0.78)	1.60 (0.79) ***	0.69 (0.57)	1.33 (0.82) ***	-	-
Vitamin B_3_, mg ^1^	10.4 (7.2)	11.8 (7.1) ***	9.9 (7.3)	9.3 (8.1) **	-	-	-	-
Vitamin B_12_, µg	2.17 (2.61)	2.75 (2.50) ***	1.88 (1.81)	2.40 (2.20) ***	2.11 (2.11)	3.04 (2.34) ***	1.08 (1.56)	1.56 (1.35) ***
Vitamin C, mg	44.0 (63.8)	77.9 (87.1) ***	21.7 (28.9)	20.7 (30.3)	11.2 (26.0)	49.8 (53.7) ***	17.8 (39.8)	18.5 (37.3)
Vitamin D, µg	3.20 (4.29)	6.75 (6.38) ***	3.82 (3.71)	6.24 (4.22) ***	1.92 (3.36)	8.50 (7.40) ***	2.18 (4.59)	4.18 (4.63) ***
Calcium, mg	467 (349)	769 (449) ***	409 (334)	691 (414) ***	309 (296)	811 (510) ***	318 (274)	512 (284) ***
Iron, mg	10.1 (6.7)	12.4 (6.8) ***	6.1 (4.0)	5.6 (4.7) **	6.3 (4.5)	10.3 (6.8) ***	6.4 (3.8)	6.7 (3.6)
Zinc, mg	-	-	4.10 (2.45)	3.99 (2.71)	4.37 (2.79)	7.08 (4.32) ***	5.32 (3.50)	5.53 (3.04)
Magnesium, mg	-	-	91.6 (61.4)	82.3 (58.4) ***	-	-	-	-
Sodium, mg	1710 (1250)	1530 (1010) ***	2010 (1400)	1810 (1270) ***	-	-	-	-
Potassium, mg	936 (582)	1190 (610) ***	1050 (657)	1110 (743) *	-	-	-	-
Phosphorus, mg	623 (386)	788 (390) ***	581 (370)	661 (431) ***	-	-	-	-
Choline, mg	-	-	-	-	195 (187)	216 (173) **	-	-
DHA, mg	-	-	-	-	30.9 (48.9)	48.8 (63.3) ***	-	-

Data are presented as medians (IQR). Data were analyzed with Wilcoxon’s rank-sum tests. Significant differences for children who did not consume dairy are indicated as *** *p* < 0.001, ** *p* < 0.01, and * *p* < 0.05. ^1^ Outliers removed in Thailand: did not consume dairy (*n* = 1842) and consumed dairy (*n* = 651).

**Table 6 nutrients-16-03229-t006:** Sociodemographic and anthropometric characteristics of children who skipped, semi-skipped, and did not skip breakfast according to the child food habit questionnaire.

	Skippers (*n* = 564)	Semi-Skippers (*n* = 471)	Non-Skippers (*n* = 9118)	*p*-Value
Age group, *n* (%)				
2–6 years (*n* = 5169)	148 (26.4)	159 (33.8)	4858 (53.3)	<0.001
7–12 years (*n* = 4984)	416 (73.8)	312 (66.2)	4260 (46.7)	
Sex, *n* (%)				
Female (*n* = 5148)	330 (58.5)	225 (47.8)	4593 (50.4)	<0.001
Male (*n* = 5005)	234 (41.5)	246 (52.2)	4525 (49.6)	
Area, *n* (%)				
Urban (*n* = 4509)	303 (53.7)	246 (52.2)	3960 (43.4)	<0.001
Rural (*n* = 5644)	261 (46.3)	225 (47.8)	5158 (56.6)	
Income group, *n* (%)				
High-income (*n* = 1639)	99 (17.6)	59 (12.5)	1481 (16.2)	<0.001
Middle-income (*n* = 2773)	96 (17.0)	90 (19.1)	2587 (28.4)	
Low-income (*n* = 3652)	227 (40.2)	174 (36.9)	3251 (35.7)	
Poor (*n* = 2043)	139 (24.6)	140 (29.7)	1764 (19.3)	
Missing values (*n* = 46)	3 (0.5)	8 (1.7)	35 (0.4)	
Country, *n* (%)				
Malaysia (*n* = 2405)	286 (50.7)	207 (43.9)	1912 (21.0)	<0.001
Thailand (*n* = 2610)	78 (13.8)	107 (22.7)	2425 (26.2)	
Indonesia (*n* = 2164)	153 (27.1)	122 (25.9)	1889 (20.7)	
Vietnam (*n* = 2974)	47 (8.3)	35 (7.4)	2892 (31.7)	
Stunted, *n* (%)				
No (*n* = 9114)	517 (91.7)	413 (87.7)	8184 (89.8)	0.11
Yes (*n* = 1039)	47 (8.3)	58 (12.3)	934 (10.2)	
Overweight/obese, *n* (%)				
No (*n* = 8155)	411 (72.9)	384 (81.5)	7360 (80.7)	<0.001
Yes (*n* = 1998)	153 (27.1)	87 (18.5)	1758 (19.3)	

Data show frequencies and proportions (%). Chi-square tests were conducted within each stratification group.

**Table 7 nutrients-16-03229-t007:** Multivariate binary logistic regression analysis of factors associated with stunting and overweight/obesity among Southeast Asian children.

Outcome and Reference	Predictor	β (SE)	95% Confidence Interval for Odds Ratio		
			Lower	Odds Ratio	Upper
**Stunting**	Intercept	−2.97 (0.13) ***	0.04	0.05	0.07
Non-skippers	Semi-skippers	0.21 (0.15)	0.91	1.23	1.63
	Skippers	−0.15 (0.16)	0.62	0.86	1.17
2–6 years old	7–12 years old	−0.40 (0.07) ***	0.59	0.67	0.77
Urban	Rural	0.23 (0.07) **	1.10	1.26	1.44
High-income group	Middle income group	0.59 (0.14) ***	1.38	1.81	2.40
	Low-income group	0.87 (0.13) ***	1.84	2.38	3.11
	Poor	1.43 (0.14) ***	3.20	4.16	5.78
**Overweight/obesity**	Intercept	−1.94 (0.08) ***	0.12	0.14	0.17
Non-skippers	Semi-skippers	−0.23 (0.13)	0.62	0.80	1.02
	Skippers	0.26 (0.10) *	1.05	1.29	1.58
2–6 years old	7–12 years old	1.17 (0.06) ***	2.90	3.23	3.60
Female	Male	0.36 (0.05) ***	1.29	1.43	1.59
High-income group	Middle income group	−0.15 (0.07) *	0.74	0.86	0.99
	Low-income group	−0.42 (0.07) ***	0.57	0.66	0.76
	Poor	−0.84 (0.09) ***	0.36	0.43	0.51

β (SE), regression coefficient (standard error). * *p* < 0.05, ** *p* < 0.01, and *** *p* < 0.001.

## Data Availability

The datasets presented in this article are not readily available because they are proprietary and managed by FrieslandCampina, which does not permit external access to the data. Requests to access the datasets should be directed to Nadja Mikulic (nadja.mikulic@frieslandcampina.com).

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
