# Peer review of "Dairy Consumption at Breakfast among Southeast Asian Children: Associations with Nutrient Intake from the South East Asian Nutrition Surveys II (SEANUTS II)"

_nutrients, 2024, doi:10.3390/nu16193229_

Round 1

Reviewer 1 Report

Comments and Suggestions for Authors

This study focuses on the relationship between dairy intake at breakfast and daily nutrient intake among Southeast Asian children. The primary conclusion highlights the essential role of including dairy in breakfast for enhancing nutrient intake and overall dietary quality, which is a self-evident conclusion, much like discovering hot water. Furthermore, while the introduction of the paper discusses the impact of skipping breakfast, frequency, and quantity on health, the actual results delve into the effects of dairy as part of breakfast on health. Therefore, the structure of the introduction should be revised.

In the introduction, the study mentions, 'Despite these benefits, a previous study in Malaysia using food habit questionnaires found that 11.7% of children aged 6 to 17 years skipped breakfast, while 12.7% did not consume breakfast daily.' However, in Table 6 of this paper, Skippers (n = 564) and Non-skippers (n = 9,118), Skippers account for only about 5%. This point needs clarification from the researchers.

What confuses me is why, after describing the differences in dairy consumption and energy intake between daily and individual breakfasts, the study uses a logistic regression model to discuss the impact of skipping breakfast on obesity and overweight. This part of the results seems disconnected from the previous section. The authors should explain why this approach was taken. Why not incorporate the model into discussions on dairy consumption and energy intake and further exploration in those areas, rather than using it in secondary results?

I find it challenging to understand the continuity in the descriptions of the following sections in the authors' results. The authors should rewrite the background to enhance readers' understanding: '3.2. Breakfast intake according to the 24-hour dietary recall, 3.3. Consumption of dairy and other food groups according to 24-hour dietary recall, 3.4. Energy and nutrient intake at breakfast between children consuming versus not consuming dairy at breakfast according to the 24-hour dietary recall, 3.5. Total daily energy and nutrient intake between children consuming versus not consuming dairy at breakfast according to the 24-hour dietary recall.'

Please ask the authors to reorganize the main core points of the paper and build a model for further discussion. Instead of stopping at identifying significant differences in a particular category in observational studies, there is a need for further discussion. Without this, the paper lacks novelty.

Regarding Table 2, more emphasis should be placed on the proportion of each item in the 'Not consumed' or 'Consumed' column, rather than horizontal percentages. Other tables should also be appropriately adjusted.

Comments on the Quality of English Language

 Minor editing of English language required.

Author Response

Thank you very much for taking the time to review this manuscript. Please find the detailed responses below and the corresponding revisions highlighted in red in the re-submitted files.

Comments 1: This study focuses on the relationship between dairy intake at breakfast and daily nutrient intake among Southeast Asian children. The primary conclusion highlights the essential role of including dairy in breakfast for enhancing nutrient intake and overall dietary quality, which is a self-evident conclusion, much like discovering hot water. Furthermore, while the introduction of the paper discusses the impact of skipping breakfast, frequency, and quantity on health, the actual results delve into the effects of dairy as part of breakfast on health. Therefore, the structure of the introduction should be revised.

Response 1: Thank you for your valuable feedback on the structure of the introduction. While the primary conclusion of our study may seem self-evident, it is important to note that there is a scarcity of research focusing on breakfast and dairy consumption among children in the South-East Asian region. Most existing studies on this topic are predominantly conducted in Western countries. Our findings highlight the significance of promoting regular breakfast consumption and incorporating dairy products into children’s diets to address nutritional deficiencies and support healthy growth and development in children in this specific demographic. Additionally, our results have the potential to inform public policies, such as the introduction and implementation of school milk programs. These programs aim to provide children with regular access to dairy products and promote balanced breakfasts, which can have a substantial impact on addressing nutritional challenges in South-East Asia. Regarding the structure of the introduction, we aimed to provide a comprehensive overview of the breakfast consumption patterns and their health benefits in this region. The second part delves into the malnutrition and nutritional intake challenges prevalent in South-East Asia, highlighting the key nutrient deficiencies that can be addressed through dairy consumption. We have revised this part to enhance clarity (page 2, lines 52-59). We then highlight how incorporating dairy into breakfast can help tackle these specific issues. We acknowledge your suggestion to revisit the structure of the introduction. However, we believe that the current structure effectively sets the stage for our study by providing necessary information and justifying the focus on dairy consumption at breakfast. We hope this clarifies our approach and rationale, and we are open to further discussion and any additional suggestions you may have.

Comments 2: In the introduction, the study mentions, 'Despite these benefits, a previous study in Malaysia using food habit questionnaires found that 11.7% of children aged 6 to 17 years skipped breakfast, while 12.7% did not consume breakfast daily.' However, in Table 6 of this paper, Skippers (n = 564) and Non-skippers (n = 9,118), Skippers account for only about 5%. This point needs clarification from the researchers.

Response 2: The two studies referenced in the introduction were conducted on children aged 6 to 17 years and 6 to 12 years, which is now clarified in the introduction (page 1-2, lines 42-46) and in the discussion (page 18, lines 655-657). In contrast, our study includes a younger age group, specifically children aged 2 to 12 years. This difference in age groups may contribute to the variation in the percentage of breakfast skippers observed. Additionally, the study cited as reference [7] classified participants into only two groups: daily breakfast consumers and non-daily breakfast consumers. In contrast, both the study referenced as [6] and our study utilized a more detailed classification, dividing participants into skippers, semi-skippers, and non-skippers. This more granular classification allows for a nuanced understanding of breakfast consumption patterns. In our study, the percentage of skippers (~5%) reflects the specific age range and classification criteria used. As shown in Table 6, children aged 2 to 6 years are less likely to skip breakfast (~26%) compared to children aged 7 to 12 years (~ 74%). This indicates that the younger age group lowers the overall prevalence of skippers. We hope this explanation clarifies the observed differences and provides a better understanding of our study’s methodology and results.

Comments 3: What confuses me is why, after describing the differences in dairy consumption and energy intake between daily and individual breakfasts, the study uses a logistic regression model to discuss the impact of skipping breakfast on obesity and overweight. This part of the results seems disconnected from the previous section. The authors should explain why this approach was taken. Why not incorporate the model into discussions on dairy consumption and energy intake and further exploration in those areas, rather than using it in secondary results?

Response 3: The primary aim of our study was to explore the relationship between dairy intake at breakfast and energy and nutrient intake both at breakfast and throughout the day among South-East Asian children. In doing so, we first described the differences in energy and nutrient intake at breakfast between children consuming versus not consuming dairy at breakfast, and throughout the day between children consuming versus not consuming dairy at breakfast. The subsequent use of a logistic regression to discuss the impact of skipping breakfast on stunting and overweight/obesity was intended to provide a comprehensive analysis of the broader implications of breakfast habits. While this may seem disconnected from the previous section, it is crucial to understand that breakfast consumption patterns, including the presence or absence of dairy, are closely linked to overall dietary habits and health outcomes. By incorporating the logistic regression model, we aimed to highlight the significant health risks associated with skipping breakfast, such as increased odds of obesity and overweight. This analysis complements our findings on dairy consumption at breakfast by emphasizing the importance of regular breakfast consumption, including dairy, in promoting healthy weight management. We chose to present these results separately to clearly delineate the different aspects of our study: the specific nutrient intake benefits of dairy at breakfast and the broader health implications of breakfast skipping. We hope this explanation clarifies our approach and rationale and we are open to any further questions or suggestions.

Comments 4: I find it challenging to understand the continuity in the descriptions of the following sections in the authors' results. The authors should rewrite the background to enhance readers' understanding: '3.2. Breakfast intake according to the 24-hour dietary recall, 3.3. Consumption of dairy and other food groups according to 24-hour dietary recall, 3.4. Energy and nutrient intake at breakfast between children consuming versus not consuming dairy at breakfast according to the 24-hour dietary recall, 3.5. Total daily energy and nutrient intake between children consuming versus not consuming dairy at breakfast according to the 24-hour dietary recall.'

Response 4: Our intention with the current structure, similar as described above with the structure of the introduction, is to provide a clear progression from general breakfast consumption (3.2. Breakfast intake according to the 24-hour dietary recall) to specific food group consumption (3.3. Consumption of dairy and other food groups according to the 24-hour dietary recall), particularly focusing on dairy, and then to analyze the impact of dairy intake on energy and nutrient intake both at breakfast (3.4. Energy and nutrient intake at breakfast between children consuming versus not consuming dairy at breakfast according to the 24-hour dietary recall) and throughout the day (3.5. Total daily energy and nutrient intake between children consuming versus not consuming dairy at breakfast according to the 24-hour dietary recall). We start with sharing results of breakfast intake based on the 24-hour dietary recall. We then detail the consumption of dairy and other food groups at breakfast to highlight dietary patterns. This is followed by a comparison of energy and nutrient intake at breakfast between children who consume dairy and those who do not, providing specific insights into the role of dairy. Finally, we analyze the total daily energy and nutrient intake between those two groups to understand the broader impact of dairy consumption at breakfast. We believe this structure allows for a logical and comprehensive exploration of the topic.

Comments 5: Please ask the authors to reorganize the main core points of the paper and build a model for further discussion. Instead of stopping at identifying significant differences in a particular category in observational studies, there is a need for further discussion. Without this, the paper lacks novelty.

Response 5: Thank you for your feedback. We understand the importance of not only identifying significant differences in observational studies but also providing a deeper discussion and developing a model for further exploration. We expanded the discussion on diary consumption by adding more details about dairy consumption in Asia (page 16, lines 573-577), associations between dairy and nutrient intake (page 16 & 17, lines 578-605), and its impact on children’s diets, growth, and development (page 17, lines 617-628). Additionally, we enhanced the discussion of our secondary objective with further insights in lines 641-642 (page 17), 648-651 (page 18), 653-657 (page 18) and 672 (page 18). Our conclusion in lines 692-711 (page 18 & 19) now emphasizes the practical implications of our study, highlighting the importance of promoting regular breakfast consumption and incorporating dairy products into children’s diets in the South-East Asian region. We also discuss the potential role of public policies, such as school milk programs, in supporting these dietary habits, which can lay the groundwork for developing a model that can be explored in future research, such as a school milk intervention study.

Comments 6: Regarding Table 2, more emphasis should be placed on the proportion of each item in the 'Not consumed' or 'Consumed' column, rather than horizontal percentages. Other tables should also be appropriately adjusted.

Response 6: We understand the suggestion to emphasize the proportion of each item in the “Not consumed” or “Consumed” column. However, our focus on horizontal percentages is intentional. By presenting the data this way, we aim to highlight the proportion of children consuming versus not consuming dairy at breakfast within specific sub-groups (e.g., how many children aged 2 to 3 years consumed dairy at breakfast). This approach allows us to better understand the distribution and characteristics of dairy consumption across different age groups, sexes, areas, and income levels. Therefore, we believe that maintaining the current presentation of proportion is crucial for the clarity and objectives of our study.

Reviewer 2 Report

Comments and Suggestions for Authors

Dear Authors,

The manuscript (ID: nutrients-3191780) submitted for review is quite interesting in the context of global public health. I propose a minor correction to the title, as the analyzes and much of the manuscript focus more on breakfast consumption in general, and not only dairy products. For example: “Breakfast consumption in South-East Asian Children: Associations with dairy consumption and h nutrient Intake from the South East Asian Nutrition Surveys II (SEANUTS II)”. Then, you should consistently change the order of keywords.

The manuscript is very interesting and is based on many results, which, thanks to well-chosen statistical analysis, allows for proper conclusions.

I have a few questions for the authors:

1.     How were the dietary interviews and questionnaires conducted among children aged 2-12? Did parents complete these questionnaires for younger and older children, or was the data collected differently?

2.     Is the lower consumption of dairy products in these countries a dietary habit related to the reluctance to consume them, or perhaps the low consumption of these products results from cultural differences among Asians? It would be good to mention this somewhere in the discussion.

3.     Why, having results for children aged six months to 12 years, was it decided only to include the age range 2 to 12 years?  Please explain.

4.     Figure 2 is not clear.

5.     Table 4 (and Table 5) is extensive; it may be worth considering using asterisks for p-values, e.g., *<0.01, **<0.001, and if the value is not statistically significant, enter ns or other abbreviation.

Limitation and Conclusion: I propose to separate sections for Limitation and Conclusion.

The Conclusion needs to be rewritten because quite enigmatic conclusions were created from such an extensive database.

References: References are not cited according to journal rules.

I believe that it concerns an important area of research in an international context.

Reviewer

Author Response

Dear Authors,

The manuscript (ID: nutrients-3191780) submitted for review is quite interesting in the context of global public health. I propose a minor correction to the title, as the analyzes and much of the manuscript focus more on breakfast consumption in general, and not only dairy products. For example: “Breakfast consumption in South-East Asian Children: Associations with dairy consumption and h nutrient Intake from the South East Asian Nutrition Surveys II (SEANUTS II)”. Then, you should consistently change the order of keywords.

The manuscript is very interesting and is based on many results, which, thanks to well-chosen statistical analysis, allows for proper conclusions.

Response General: Thank you very much for tacking the time to review this manuscript. Please find the detailed responses below and the corresponding revisions highlighted in red in the re-submitted files. Thank you for acknowledging the importance of our study in the context of global public health. We appreciate your suggestion regarding the title correction. However, we respectfully disagree with the proposed change. Our manuscript primarily focuses on dairy consumption at breakfast, examining different socio-economic groups and the differences in nutrient intake between children consuming versus not consuming dairy at breakfast, which is our primary objective. While we do touch upon breakfast consumption in general, this is mainly to provide information on different food group consumption at breakfast and to explore associations between breakfast skipping and stunting or overweight/obesity, which is our secondary objective. We believe that changing the title as suggested might lead readers to expect a broader analyses of breakfast consumption in regards of other food groups, whereas our study is specifically centered on dairy consumption. We hope our perspective is understandable and is agreeable with our decision.

I have a few questions for the authors:

Comments 1:     How were the dietary interviews and questionnaires conducted among children aged 2-12? Did parents complete these questionnaires for younger and older children, or was the data collected differently?

Response 1: To avoid redundancy in the article, we referred to our other paper, which details the methodology of the one-day 24-hour dietary recall and the one-week child food habit questionnaire (CFHQ). To address the question, the 24-hour dietary recall, which covers the period from 12 AM to 12 AM the following day and includes both weekdays and weekends, and the CFHQ, comprising of multiple-choice and open-end questions, were conducted through face-to-face or telephone interviews. For children under 10 years old, caregivers provided the information. For children aged 10 years and above, the children themselves provided the information, except in Thailand, where caregivers reported for children of all age groups.  

Comment 2:     Is the lower consumption of dairy products in these countries a dietary habit related to the reluctance to consume them, or perhaps the low consumption of these products results from cultural differences among Asians? It would be good to mention this somewhere in the discussion.

Response 2: We appreciate your suggestion to discuss the reasons behind the lower consumption of dairy products in Asian countries. In response, we have added a paragraph in the discussion section (page 16, lines 573-577). This addition aims to highlight the cultural and economic factors contributing to the lower dairy consumption in these regions.

Comment 3:     Why, having results for children aged six months to 12 years, was it decided only to include the age range 2 to 12 years?  Please explain.

Response 3: Thank you for highlighting this point. The reason for including only children aged 2 to 12 years is that the child food habit questionnaire was administered only to children aged 2 years and above. To maintain consistency, we included the same age group for both objectives. This information has been added to the methodology section (page2, lines 89-91). Additionally, we felt that collecting comprehensive data on food intake across all food groups would be incomplete for children younger than 2 years old.

Comments 4:     Figure 2 is not clear.

Response 4: We have made several adjustments to Figure 2 (page 8, line 281) to enhance its clarity. The layout of the panel structure has been changed from a 2x2 panel to a linear 1x4 panel arrangement. This adjustment aims to better distinguish the data for each country, additionally also highlighting the boxes for different countries more prominently. The y-axis label has been updated to “Number of dairy servings” to clearly indicate the type of servings being measured, replacing the previous “Number of serving” label. The x-axis label “Throughout the day” has been changed to “Total daily intake” to more accurately reflect the data presented. Further, a new color palette has been introduced for each grouping factor to facilitate easier differentiation between them. We have also revised the figure caption to improve clarity (page 9, lines 284 & 287-288). Additionally, corrections have been made to the significant bars in the graphic (page 9, lines 298-300) and the serving size of dairy in the three age groups in Thailand, which was a rounding issue (page 7, line 275), and Indonesia (page 8, line 280). We identified and corrected the error in our previous analysis. Specifically, there is no significant difference in dairy intake at breakfast between urban and rural areas in Vietnam. However, there is a significant difference in total daily dairy intake between the poverty and low-income group in Thailand. We corrected these in the figure and in the respective results sections.

Comments 5:     Table 4 (and Table 5) is extensive; it may be worth considering using asterisks for p-values, e.g., *<0.01, **<0.001, and if the value is not statistically significant, enter ns or other abbreviation.

Response 5: Thank you for your valuable suggestion. We agree that using asterisks to indicate p-values is a good idea. Consequently, we have removed the p-value columns and used asterisks to denote significance in Table 4 (pages 9 & 10), Table 5 (page 12), and the Supplementary Tables S1-S8 in the supplementary material. We have also added explanatory sentences to the table captions for table 4 (page 10, lines 327-329) and for table 5 (page 12, lines 418-420). The same changes have been applied to the supplementary material in Tables S1-S8.

Comments 6: Limitation and Conclusion: I propose to separate sections for Limitation and Conclusion.

The Conclusion needs to be rewritten because quite enigmatic conclusions were created from such an extensive database.

Response 6: We agree with your suggestion and have created a separate Conclusion section, which we have also rewritten to provide a more comprehensive summary of our findings (page 18 & 19, lines 692-711). Additionally, we have specified the key nutrients with higher intakes in the first paragraph of the discussion (page 16, lines 557-558), and incorporated this information into the conclusion.

Comments 7: References: References are not cited according to journal rules.

Response 7: Thank you for pointing this out. We have revised the references to comply with the journal’s citation guidelines. Additionally, we have addressed the self-citation rate, ensuring it is now below 15%.

I believe that it concerns an important area of research in an international context.

Reviewer 3 Report

Comments and Suggestions for Authors

Very interesting study. The authors provide very important information regarding the importance of dairy products to improve the nutritional quality of children's breakfast. The methodology used is adequate and sufficient. The results support the discussion. However, I have the following comments.

I. Comments:

1. Improve the wording of the objective of the study.

2. Figure 1: Increase the size of the letters and numbers. It was not easy to read and understand all the information.

3. Tables 4 and 5 provide very important information. Considering the most important nutrients provided by dairy products, I suggest making an additional figure that can be included in the discussion. Highlighting the contribution of proteins, calcium, phosphorus and zinc.

4. Tables 4 and 5, does DHA correspond to docosahexaenoic acid (C22:6n-3, DHA)? This is an n-3 PUFA that is not found in dairy products (naturally). So, is it incorporated in dairy products in Indonesia? If so, I suggest providing that information, and briefly discussing that benefit. Especially for the effects of DHA on neurodevelopment.

5. The inclusion of dairy products in children's diet is relevant to improve the nutritional quality of the diet. Discuss this point.

6. The inclusion of dairy products in the diet is important to prevent childhood malnutrition and obesity. As well as to promote adequate growth and development. Discuss this point.

7. What projections would this study have? For example, proposing public policies on food, nutrition and child health?

8. What could the authors discuss regarding recommendations from some researchers who suggest that dairy products are not necessary in children's breakfast?

Author Response

Very interesting study. The authors provide very important information regarding the importance of dairy products to improve the nutritional quality of children's breakfast. The methodology used is adequate and sufficient. The results support the discussion. However, I have the following comments.

Response General: Thank you very much for taking the time to review this manuscript and acknowledging the importance of our study. We greatly appreciate your comments and suggestions and believe that the additions you recommended have significantly enhanced the value of our article, particularly in the discussion section. Please find the detailed responses below and the corresponding revisions highlighted in red in the re-submitted files.

Comments 1: Improve the wording of the objective of the study.

Response 1: We agree and have revised the wording of the study’s objectives to improve clarity and precision. Specifically, we modified the first objective to explicitly state that the associations between dairy consumption at breakfast and energy and nutrient intake were assessed both at breakfast and throughout the day in the abstract (page 1, line 21), in the introduction (page 2, lines 70-72), and in the discussion (page 16, line 548). This change provides a clearer distinction between the timing of nutrient intake assessment. We also retained the secondary objective of examining associations between breakfast consumption and the prevalence of stunting and overweight/obesity, while emphasizing the limited prior research on these topics in Southeast Asia in the introduction (page 2, lines 73-74), and in the discussion (page 16, line 550).

Comments 2: Figure 1: Increase the size of the letters and numbers. It was not easy to read and understand all the information.

Response 2: We appreciate the feedback regarding Figure 1. Initially, when creating the figure, the font size was set to 12 pt. Instead of increasing it further, we have simplified the figure by removing the information about the total SEANUTS II population (n = 13,933), as the countries are analyzed separately for the primary objective (page 5, line 199). Additionally, we have consolidated the exclusion criteria, including children aged below two years, into a single exclusion box for each country. To further improve clarity, we have color-coded the primary and secondary objectives. We also increased the figure size in the manuscript to extend beyond the text narrows. Furthermore, we have slightly changed the figure text (page 5, lines 202-203) to improve clarity.

Comments 3: Tables 4 and 5 provide very important information. Considering the most important nutrients provided by dairy products, I suggest making an additional figure that can be included in the discussion. Highlighting the contribution of proteins, calcium, phosphorus and zinc.

Response 3: Thank you for the valuable suggestion. We agree that highlighting the contribution of key nutrients is important. However, given the length of the article and the extensive information already included, we aim to keep the manuscript concise and avoid redundancy. Adding multiple figures for different nutrients would require separate y-axis dimensions (g, mg, µg), which would complicate the presentation and add more figures. Nevertheless, we agree to emphasize the contributions of key nutrients in the discussion. We have added additional information on page 16 & 17, lines 578-605 to address this.

Comments 4: Tables 4 and 5, does DHA correspond to docosahexaenoic acid (C22:6n-3, DHA)? This is an n-3 PUFA that is not found in dairy products (naturally). So, is it incorporated in dairy products in Indonesia? If so, I suggest providing that information, and briefly discussing that benefit. Especially for the effects of DHA on neurodevelopment.

Response 4: Indeed, DHA corresponds to docosahexaenoic acid, which is not naturally found in dairy products. However, some milk products in South-East Asia, including Indonesia, are fortified with DHA. We included DHA intake data specifically from Indonesia, as reliable DHA information was not available in the local food composition databases of the other countries. Our nutrient intake data reflect general dietary intake and not limited to nutrients naturally present in dairy. We recognize that many nutrients, including DHA, are present in dairy products primarily through fortification. Therefore, we included as many nutrients as possible from the food composition databases. However, we did not delve deeply into fortification-specific nutrients, including DHA, as it was only available for one country.

Comments 5: The inclusion of dairy products in children's diet is relevant to improve the nutritional quality of the diet. Discuss this point.

Response 5: We agree and have addressed this point in the discussion section (page 17, lines 617-621), where we discuss how including dairy products into children’s diet improves diet quality. Additionally, we would like to point out to the paragraph we already had included, describing the health benefits of replacing carbohydrate-rich breakfast components with one serving of dairy (page 17, lines 628-633).

Comments 6: The inclusion of dairy products in the diet is important to prevent childhood malnutrition and obesity. As well as to promote adequate growth and development. Discuss this point.

Response 6: Our study primarily analyzes the associations between breakfast skipping and stunting and overweight/obesity, rather than directly examining the impact of dairy intake on preventing childhood malnutrition and overweight/obesity. However, while our focus is on the effects of dairy intake at breakfast on nutrient intake and the associations of breakfast skipping and stunting and overweight, we contextualized our findings within the broader literature on dairy consumption and childhood anthropometric outcomes in the discussion (page 17, lines 621-628).

Comments 7: What projections would this study have? For example, proposing public policies on food, nutrition and child health?

Response 7: The findings of this study have significant implications for public policies. Promoting dairy consumption, especially at breakfast, can help ensure that children receive essential nutrients such as calcium and vitamin D, which are crucial for growth and development. These nutrients are insufficiently consumed in this region, as shown in our previous research from SEANUTS II. Incorporating dairy into the diet can improve overall diet quality and nutrient intake. Given the association between skipping breakfast and an increased risk of overweight and obesity, public health campaigns should emphasize the importance of a nutritious breakfast that includes dairy. Implementing school milk programs can further support this goal, potentially reducing the prevalence of malnutrition and obesity. We emphasized this by adding further information to the conclusion (page 19, lines 700-708).

Comments 8: What could the authors discuss regarding recommendations from some researchers who suggest that dairy products are not necessary in children's breakfast?

Response 8: Several studies highlight the importance of dairy in contributing to nutrient intake in children. Our study also shows that many children do not consume dairy foods according to local recommendations, and encouraging dairy consumption can be beneficial. We acknowledge that some researchers suggest dairy products are not necessary for children’s breakfast, often due to concerns about environmental impacts or lactose intolerance. While these are valid points, discussing them in detail would exceed the scope of our article.

Round 2

Reviewer 1 Report

Comments and Suggestions for Authors

Thanks to the author's response, the changes and explanations in the above revision culture have mostly replied to my question

Comments on the Quality of English Language

 Minor editing of English language required.